# Construction of Recombinant Rabies Virus Vectors Expressing H or F Protein of Peste des Petits Ruminants Virus

**DOI:** 10.3390/vetsci9100555

**Published:** 2022-10-10

**Authors:** Haojie Wang, Jinhao Bi, Na Feng, Yongkun Zhao, Tiecheng Wang, Yuetao Li, Feihu Yan, Songtao Yang, Xianzhu Xia

**Affiliations:** 1Changchun Veterinary Research Institute, Chinese Academy of Agriculture Sciences, Changchun 130000, China; 2College of Animal Science and Veterinary Medicine, Henan Institute of Science and Technology, Xinxiang 130118, China; 3College of Veterinary Medicine, Jilin Agriculture University, Changchun 453003, China

**Keywords:** peste des petits ruminants, rabies virus vector, reverse genetics, vaccine candidate strain, bivalent vaccine

## Abstract

**Simple Summary:**

Peste des petits ruminants (PPR) is one of the most contagious and fatal diseases of small ruminants. In this study, two recombinant viruses rSRV9-H and rSRV9-F, which express the envelope glycoprotein H (hemagglutinin protein) or F (fusion protein) protein, respectively, were successfully generated with a rabies virus as vector. The constructed viruses had good proliferative activity and stability and provided potential bivalent inactivated vaccine candidate strains for the prevention of PPR and livestock rabies.

**Abstract:**

Peste des petits ruminants (PPR) is one of the most contagious and fatal diseases of small ruminants in the world and is classified as a category A epidemic disease. It is the target of a global eradication campaign led by the Office International des Epizooties (OIE) and Food and Agriculture Organization of the United Nations (FAO). The PPR live attenuated vaccine is currently the most widely used and approved vaccine, but the use of this vaccine interferes with the serological testing of the PPR elimination program, and there is a potential safety risk. Viral vector vaccines are one of the most promising methods to solve this dilemma. In this study, the full-length infectious clone plasmid of rabies virus (RABV), pD-SRV9-PM-LASV, was used as the backbone, and the envelope glycoprotein H (hemagglutinin protein) or F (fusion protein) gene of PPRV was inserted into the backbone plasmid to construct the infectious clones pD-SRV9-PM-PPRV-H and pD-SRV9-PM-PPRV-F, which express the PPRV H and PPRV F genes, respectively. The correct construction of these infectious clones was verified after sequencing and double digestion. The infectious clones were transfected with a helper plasmid into BSR/T7 cells, and recombinant viruses were successfully rescued by direct immunofluorescence, indirect immunofluorescence, Western blotting, and transmission electron microscopy and named rSRV9-H and rSRV9-F. The results of growth kinetics studies indicated that the inserted gene did not affect virus proliferation. Stability studies revealed that the inserted target gene was stably expressed in recombinant RABV for at least 15 generations. In this study, the recombinant viruses rSRV9-H and rSRV9-F were successfully rescued. The constructed viruses had good proliferative activity and stability and provided potential bivalent inactivated vaccine candidate strains for the prevention of PPR and livestock rabies.

## 1. Introduction

Peste des petits ruminants (PPR) is a viral disease, caused by the small ruminant morbillivirus (SRMV, commonly termed as PPRV), a member of the Morbillivirus genus, in the family Paramyxoviridae [1]. Based on genetic relationships between PPR viruses from different geographical regions, PPRVs have undergone independent evolution which has resulted in four genetic subtypes (PPRV lineages I–IV) [1,2]. PPR is one of the most contagious and fetal diseases of small ruminants in the world. The Office International des Epizooties (OIE), in the *Terrestrial Animal Health Code*, classifies PPR as a category A disease in the infection of goats and sheep [3]. PPR is also widely spread among wild ruminants and can spread across international borders. It is clinically and pathologically like rinderpest and human measles, and its high morbidity and mortality cause devastating economic losses to the global animal husbandry industry. The existence of PPR, especially in developing countries in Africa, Asia, and Europe, will continue to reduce food security, threaten the livelihood of and trade by herders, and negatively impact biodiversity and ecosystem health. PPR is currently a target of a global eradication campaign led by the OIE and Food and Agriculture Organization of the United Nations (FAO) [4].

The current method to prevent PPR is vaccination with a live attenuated homologous PPRV vaccine, such as Nigeria 75/1, which belongs to lineage II, and three vaccines developed from the lineage IV virus, namely Sungri/96, Arasur/87, and Coimbatore/97 [5,6]. Although these live attenuated vaccines have good immunization effects, they have low heat resistance. In addition to poor heat resistance, there is also a risk of virulence recovery and the inability to differentiate infected from vaccinated animals (DIVA), which are very important for seroepidemiological monitoring after eradication and confirmation of PPR-free areas [7]. This shortcoming can be overcome using a vaccine capable of DIVA and its corresponding serological test. Viral vector vaccines are also suitable for the DIVA strategy because a vector virus only expresses one or two proteins of the corresponding wild-type virus. A recombinant vector virus that expresses the PPRV immunogenic surface protein F or H can provide protection against PPRV infection by inducing PPRV neutralizing antibodies. The current viral vector vaccines mainly use goat pox virus [8,9], Newcastle disease virus [10], adenovirus [11], and other viruses as the vectors. However, to date, such vaccines have not been approved for use.

Rabies virus (RABV) has obvious advantages as a vaccine vector delivery platform. First, RABV can replicate and transcribe efficiently in target cells. In addition, because few animals have anti-RABV serum antibodies, the preexisting RABV seropositivity rate is negligible. Therefore, using RABV as the backbone to construct a recombinant RABV vector vaccine against PPRV is feasible and has the potential to prevent and control RABV. The RABV has indeed been used as a vector in vaccine fields. The RABV vector has been successfully used in developing vaccines against severe fever with thrombocytopenia syndrome (SFTS) [12], bovine ephemeral fever virus [13], and Nipah virus [14]. Our lab has also focused on RABV vector studies for years, and has developed a Rift Valley Fever Virus Vaccine with this system [15].

Therefore, in this study, the full-length infectious clone plasmid of RABV, pD-SRV9-PM-LASV, was used as the backbone in which to insert the PPRV envelope glycoprotein H (hemagglutinin protein) or F (fusion protein) gene. The recombinant rabies viruses rSRV9-H and rSRV9-F were rescued through reverse genetics, and the proliferation activity and stability of the recombinant viruses were verified, providing potential vaccine candidate strains for the prevention of PPR and livestock rabies.

## 2. Materials and Methods

### 2.1. Viruses, Cells, Serum, and Plasmids

BSR/T7 cells; recombinant full-length infectious clone pD-SRV9-PM-LASV derived from the SRV9 strain (GenBank Accession: AF499686); helper plasmids pcDNA3.1-SRV9-N, pcDNA3.1-SRV9-P, and pcDNA3.1-SRV9; and sera containing anti-PPRV H and PPRV F protein polyclonal antibodies were all preserved and obtained from the Laboratory of Animal Virology and Special Animal Epidemiology, Changchun Veterinary Research.

### 2.2. Target Gene Synthesis

The CDS regions of the H and F genes were synthesized using the coding sequences (CDSs) of the H and F proteins of the PPRV China/Tibet/Geg/07-30 (GenBank FJ905304.1) strain published in GenBank. The CDS regions of H and F were ligated into the pUC57 vector through the enzyme cleavage sites PstI and KpnI; the vectors were named pUC57-PPRV-H and pUC57-PPRV-F, respectively. Gene synthesis was completed by Sangon Biotech (Shanghai, China).

### 2.3. Amplification and Purification of the Target Gene

A total of 40 μL of sterile enzyme-free water was added to the tubes containing lyophilized pUC57-PPRV-H and pUC57-PPRV-F and centrifuged, and 1 μL of each plasmid was used to transform HST08 competent cells. Using the pUC57-PPRV-H and pUC57-PPRV-F plasmids as templates, the target genes were amplified using the primers listed in Table 1. The reaction system was as follows: pUC57-PPRV-H/pUC57-PPRV-F1 forward primer (10 μM), 1.5 μL; reverse primer (10 μM), 1.5 μL; Prime STAR^®^Max DNA Polymerase (2×), 25 μL; and ddH_2_O, 21 μL. The reaction program was as follows: predenaturation at 98 °C for 1 min, followed by 4 cycles of 10 s at 98 °C, 5 s at 60 °C, 10 s at 72 °C (5 s/kb), and 10 min at 72 °C. Target products were recovered by gel purification and named PPRV-H-PCR and PPRV-F-PCR.

### 2.4. Construction and Verification of Recombinant Full-Length Infectious Clones

Using the RABV/SRV9 reverse genetic operating system established in our laboratory [15], 2 recombinant SRV9 infectious clones expressing PPRV H and PPRV F were constructed (Figure 1a). The recombinant SRV9 infectious clone plasmids pD-SRV9-sPM-PPRV-H and pD-SRV9-sPM-PPRV-F were constructed by replacing the eGFP gene (Figure 1a) with the PPRV H and PPRV F genes through the BsiWI and PmeI restriction sites, respectively. The details are as follows: (1) pD-SRV9-PM-LASV and the plasmids PPRV-H-PCR and PPRV-F-PCR recovered by gel purification were digested with BsiWI and PmeI at 37 °C. The digested products were electrophoresed. The SRV9 vector and the target gene fragments of PPRV H and PPRV F were recovered by gel purification. (2) The amount of each component of the recovered SRV9 vector and the 2 target fragments was calculated using a molar ratio of 1:3, 5 μL of (4×) Anza™ T4 DNA ligase master mix was added (total volume, 20 μL), and the system was incubated at room temperature overnight. After ligation, the products were transformed into HST08 competent cells, spread onto culture dishes containing bacterial solid medium, and incubated overnight. (3) Single clones were selected for preliminary identification by PCR, and the correct plasmids were further identified by digestion with the restriction endonucleases BsiWI and PmeI. Plasmids with the correct insert were sent to Comate Bioscience (Jilin, China) for sequencing. Plasmids with the correct sequences were named pD-SRV9-PM-PPRV-H and pD-SRV9-PM-PPRV-F.

### 2.5. Rescue of Recombinant Virus

The full-length infectious clone pD-SRV9-PM-PPRV-H/F (2.5 μg) and the helper plasmids pcDNA3.1-RABV N (0.625 μg), pcDNA3.1-RABV P (0.3125 μg), pcDNA3.1-RABV L (0.125 μg), and pcDNA3.1-RABV G (3.7875 μg) were transfected into BSR/T7 cells using Lipo2000 transfection reagent to rescue the recombinant virus. After transfection, the cells were cultured in an incubator for 4–5 h, the liquid in the wells was aspirated and discarded, and 2 mL of DMEM with 10% FBS was added to each well, after which the cells were again incubated. At 3 days after transfection, 1 mL of cell supernatant was aspirated from each well, labeled, and stored at −80 °C for identification purposes. Each well was supplemented with 1 mL of DMEM containing 10% FBS. Seven days after transfection, the 6-well plates were sealed with adhesive tape and frozen at −80 °C and thawed. The cell solution in each well was mixed by pipetting in a virus handling biosafety cabinet. The mixture of cell debris and cell solution was aliquoted and named rSRV9-H and rSRV9-F.

### 2.6. Direct Immunofluorescence and Indirect Immunofluorescence

The night before virus inoculation, BSR/T7 cells were passaged into 48-well cell culture plates. The culture medium was discarded, the cells were washed, and 200 μL of harvested P1 virus was inoculated into each well. Normal cells were used as a negative control. Direct immunofluorescence detection was performed using a FITC-labeled mouse anti-RABV N protein monoclonal antibody (FDI FUJIREBIO) at a 1:200-fold dilution. Rabbit anti-PPRV H and PPRV F polyclonal antibody serum and a FITC-labeled goat anti-rabbit IgG antibody (Sigma, St. Louis, MO, USA)) were used for indirect immunofluorescence detection.

### 2.7. Western Blot

To determine whether the PPRV H and PPRV F inserts in the recombinant virus successfully expressed rSRV9-H and rSRV9-F, the P3, P6, P9, P12, P15, and P18 generations of recombinant RABV containing rSRV9-H and rSRV9-F were analyzed. A 50-fold dilution of rabbit PPRV H and PPRV F polyclonal antibodies was used as the primary antibody, and HRP-labeled goat anti-rabbit IgG antibody (dilution, 1:20,000) was used as the secondary antibody for Western blot identification.

### 2.8. Transmission Electron Microscopy (TEM)

The morphology and size of the recombinant virus were observed by TEM. Recombinant virus supernatant inactivated with glutaraldehyde was added dropwise onto a special copper mesh for electron microscopy; the supernatant was incubated at room temperature for 5 min and then stained with 1% phosphotungstic acid for 3 min. The excess liquid on the edge of the copper mesh was removed using filter paper. After the copper mesh was dry, observation was performed under a transmission electron microscope.

### 2.9. Growth Kinetics of Recombinant RABV

Recombinant viruses with stable virulence were used for the growth kinetics analysis. Based on the equation $x=\frac{\mathrm{MOI}\times \mathrm{cell}\;\mathrm{numbers}}{\mathrm{TCID}50}$, MOI = 0.1 was used to calculate the corresponding inoculation volume. Using the calculated inoculation volume, recombinant RABV rSRV9-H, rSRV9-F, and SRV9 were inoculated into T25 culture flasks, incubated at 37 °C for 1 h, and cultured in a 37 °C incubator with DMEM containing 2% FBS. At 0, 24, 48, 72, and 96 h, 200 μL of supernatant was collected. The TCID_50_ (median tissue culture infectious dose) of the viruses at the 5 timepoints was determined using the TCID_50_ method.

### 2.10. Genetic Stability of Antigenic Genes in Recombinant RABV

RNA was extracted from the P3 generation, P6 generation, P9 generation, P12 generation, and P15 generation of recombinant RABV, rSRV9-H, and rSRV9-F using a TIANamp virus RNA extraction kit (Tiangen, Beijing, China). After transcribing the RNA into cDNA using TransScript One-Sep RT–PCR SuperMix (TransGen, Beijing, China), identification was performed via PCR using the primers listed in Table 1. The sizes of the bands were analyzed, and gel blocks containing DNA of the correct size were sent to Comate Bioscience (Jilin, China) for sequencing.

## 3. Results

### 3.1. Construction and Identification of pD-SRV9-PM-PPRV-H and pD-SRV9-PM-PPRV-F

Using the RABV/SRV9 reverse genetic operating system established in our laboratory (Figure 1a), the PPRV H and PPRV F genes were used to replace the eGFP gene (Figure 1a) through two enzyme cleavage sites, BsiWI and PmeI, to construct recombinant infectious clones. To determine whether the infectious clones pD-SRV9-PM-PPRV-H and pD-SRV9-PM-PPRV-F were successfully constructed, they were digested and identified using restriction enzymes. The actual sizes of the SRV9 vector (17,049 bp) (Figure 1c), PPRV H gene (1841 bp) (Figure 1a), and F gene (1640 bp) (Figure 1b) were consistent with expectations. The successful construction of the two clones was further verified by sequencing.

### 3.2. Direct Immunofluorescence Identification of the Successful Rescue of the Recombinant Viruses

Green fluorescence was not detected in the negative control BSR/T7 cells without recombinant virus inoculation (Figure 2b,d), and cluster-specific green fluorescence was observed in the recombinant viruses, rSRV9-H (Figure 2a) and rSRV9-F (Figure 2c), which confirmed the expression of the RABV N protein and further proved that rSRV9-H and rSRV9-F were successfully rescued.

### 3.3. Identification of Recombinant Virus Rescue via Indirect Immunofluorescence

The expression of PPRV envelope glycoproteins H and F in the recombinant viruses rSRV9-H and rSRV9-F was detected by indirect immunofluorescence. The results showed that specific green fluorescence was detected in the recombinant viruses rSRV9-H (Figure 3a) and rSRV9-F (Figure 3c) under a fluorescence microscope, while the negative control showed no fluorescence (Figure 3b,d). The results indicated that the rescued rSRV9-H and rSRV9-F successfully expressed two exogenous envelope glycoproteins.

### 3.4. Verification of the Correct Expression of Recombinant Virus H and F via Western Blot

The recombinant viruses rSRV9-H and rSRV9-F were harvested and inactivated with β-propiolactone. To verify the expression of PPRV H and PPRV F, Western blot analysis was performed. Using PPRV H and PPRV F rabbit polyclonal antibodies as primary antibodies, 70 kDa PPRV H (Figure 4a) and 60 kDa PPRV F (Figure 4b) were detected in the recombinant viruses rSRV9-H and rSRV9-F, confirming PPRV H protein and PPRV F protein expression in the recombinant virus.

### 3.5. TEM Revealed That the Recombinant Virus Particles Were Bullet-Shaped

The supernatants of the two recombinant virus cultures were inactivated with β-propiolactone and observed under a transmission electron microscope. The recombinant viruses rSRV9-H (Figure 5a) and rSRV9-F (Figure 5b) were found to be bullet-shaped.

### 3.6. Growth Kinetics of Recombinant Viruses

To examine whether the recombinant PPRV H and PPRV F genes of RABV SRV9 affected the growth kinetics of the virus, the growth kinetics of the recombinant viruses rSRV9-H and rSRV9-F and the parental virus SRV9 were compared in BSR/T7 cells. The results showed that the growth of the three viruses peaked at 72 h after inoculation, similar to that for the parental virus, further indicating that the growth kinetics of the virus were not affected after viral recombination (Figure 6).

### 3.7. Stable Inheritance of Exogenous Gene Expression in Recombinant Viruses

The RNA genomes of the P3, P6, P9, P12, and P15 generations of recombinant viruses were extracted for RT–PCR; the results are shown in Figure 7. The corresponding antigen genes were detected in the genomes of the recombinant viruses of each generation, and the PCR products were verified by sequencing. The results showed that the inserted gene was stably inherited and correctly expressed in the recombinant RABV.

## 4. Discussion

RABV has advantages as a vaccine vector. It has a relatively simple genome, which allows easy manipulation of cDNA through traditional cloning techniques [16]. For example, exogenous genes such as vaccine antigen target genes can be stably integrated into the RABV genome [17,18]. Based on the transcriptional pattern of RABV, the optimal position for the maximum expression of exogenous genes is near the 3’ end of the RABV genome. However, the insertion of exogenous genes at the proximal end of the promoter may more profoundly interfere with viral replication than insertion at the distal end of the promoter [19]. In addition, the viral genome can maintain replication after the insertion of multiple large genes [20]. Moreover, compared with other viral vectors, such as adenovirus, the presence of RABV serum antibodies in the general population is negligible [21,22]. In addition, in recent years, cases of RABV infection and death have frequently occurred in sheep in pastoral areas due to attacks by wild animals such as foxes [23]. The prevention of rabies has become an important topic in the free-range animal husbandry industry. Therefore, in this study, RABV was used as the backbone to prepare a PPRV and RABV recombinant virus, and vaccine candidate strains for PPRV and RABV prevention and control were optimized.

The RABV genome is separated by the conserved regulatory region of recombinant RABV, which is also called the gene boundary. This regulatory region includes the termination signal at the gene terminus and the gene initiation signal. By introducing this gene boundary into the recombinant RABV genome, additional genes can be introduced to express proteins, which is why rhabdoviruses such as RABV are suitable as gene expression vectors [24]. The transcription of RABV mRNA gradually decreases from the 3’ end to the 5’ end, which results in a transcription gradient in the gene sequence. Therefore, theoretically, gene expression can be enhanced by moving the position of the inserted transcription unit closer to the 3’ end of the genome. Studies have shown that the addition of additional transcription units between the N and P genes in the RABV/SRV9 genome attenuates viral replication [25]. We previously inserted exogenous genes such as eGFP (720 bp) and MERS_S1_ (2408 bp) between the P and M of the RABV genome and successfully rescued the recombinant virus and expressed the exogenous protein [26]. Therefore, the insertion of the PPRV H and PPRV F genes at the same position can theoretically successfully rescue the virus and achieve the effective expression of the exogenous protein.

Most of the immune-related protective determinants of the measles virus have been localized to F, H, and N. A large number of studies have shown that the H and F encoded by PPRV are sufficient to induce protective immune responses in infected animals [27,28,29]. In addition, some studies have shown that the immunogenicity of the PPRV lineage IV strain Tibet/30 is higher than that of the lineage II virus strain Nigeria75/1 [30]. Therefore, in this study, the F and H coding regions of the Tibet/30 strain were used as the target genes and inserted into the RABV/SRV9 genome to express the fusion protein and hemagglutinin protein.

In this study, the pD-SRV9-PM-PPRV-H and pD-SRV9-PM-PPRV-F full-length plasmids were constructed. After the two full-length plasmids and helper plasmids were transfected into BSR/T7 cells, direct immunofluorescence and indirect immunofluorescence were used to verify that the two recombinant viruses were successfully rescued. Western blotting and RT–PCR demonstrated that the recombinant virus stably expressed exogenous proteins and genes. Our study provides potential vaccine candidate strains for the prevention of PPR and livestock rabies. For safety reasons, in subsequent vaccine development, we will inactivate the recombinant viruses rSRV9-H and rSRV9-F and conduct subsequent inactivated vaccine-related studies. We will also evaluate the immune efficacy of the peste des petits ruminants recombinant rabies virus vaccines with a mouse model.

## 5. Conclusions

In this study, the recombinant viruses rSRV9-H and rSRV9-F were successfully rescued. The constructed viruses had good proliferative activity and stability and provided potential bivalent inactivated vaccine candidate strains for the prevention of PPR and livestock rabies.

## Figures and Tables

**Figure 1 vetsci-09-00555-f001:**
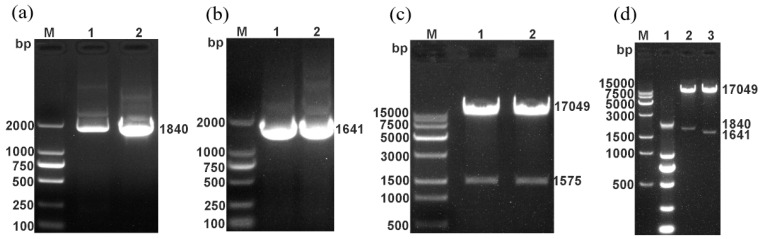
Identification of pD-SRV9-PM-H and pD-SRV9-PM-F via enzyme digestion. (**a**) The construction strategy for the recombinant RABV infectious clone. (**b**–**d**) Identification of pD-SRV9-PM-H and pD-SRV9-PM-F via enzyme digestion. (**a**) M: DNA Marker; 1,2: PPRV-H-PCR; (**b**) 1, 2: PPRV-F-PCR; (**c**) DNA Marker; 1, 2: digestion of pD-SRV9-PM-LASV; (**d**) 2: identification of pD-SRV9-PM-PPRV-H via enzyme digestion; 3: identification of pD-SRV9-PM-PPRV-F via enzyme digestion.

**Figure 2 vetsci-09-00555-f002:**
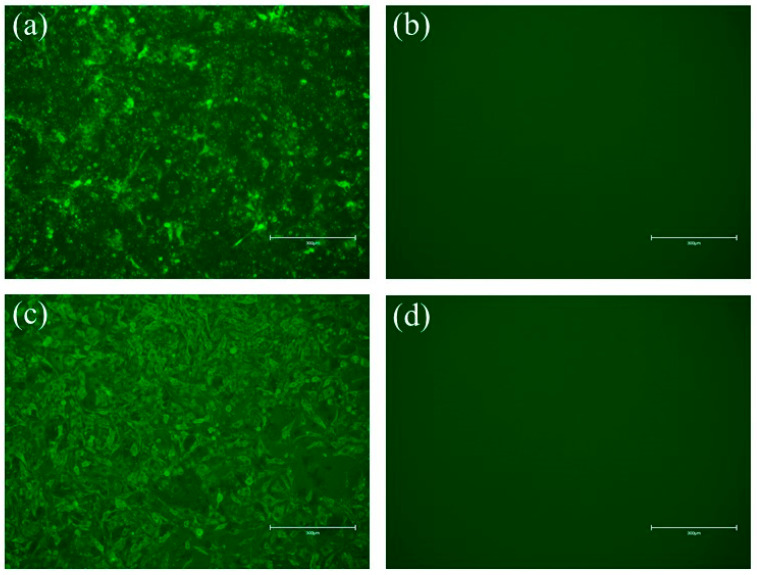
Identification of the recombinant viruses rSRV9-H and rSRV9-F via direct immunofluorescence. Cells were labeled with FITC-labeled mouse anti-RABV N protein monoclonal antibody at a 1:200-fold dilution. (**a**) Recombinant virus rSRV9-H, (**b**) negative control, (**c**) recombinant virus rSRV9-F, (**d**) negative control. Observation was performed using a fluorescence microscope.

**Figure 3 vetsci-09-00555-f003:**
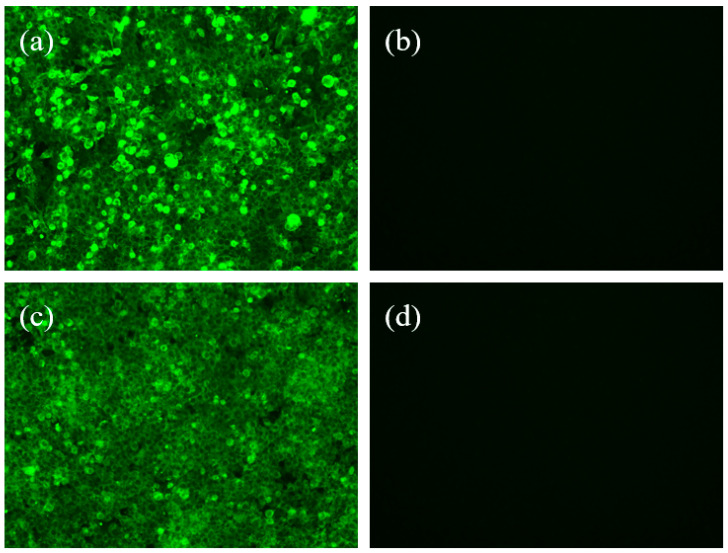
Identification of recombinant viruses rSRV9-H and rSRV9-F by indirect immunofluorescence. The viruses were labeled with rabbit anti-PPRV H and PPRV F polyclonal antibody serum and an FITC-labeled goat anti-rabbit IgG antibody and observed under a fluorescence microscope. (**a**) Recombinant virus rSRV9-H, (**b**) negative control, (**c**) recombinant virus rSRV9-F, (**d**) negative control.

**Figure 4 vetsci-09-00555-f004:**
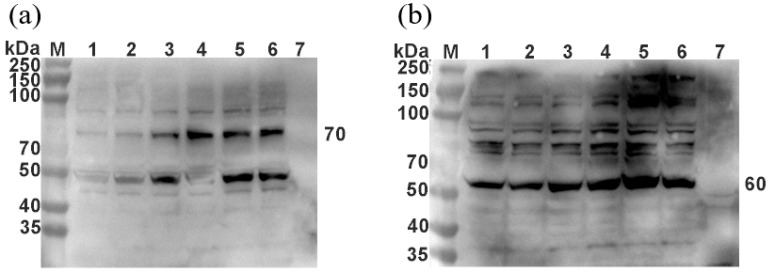
Identification of the recombinant viruses rSRV9-H and rSRV9-F via Western blot. The P3, P6, P9, P12, P15, and P18 generations of recombinant RABV rSRV9-H and rSRV9-F were collected and subjected to Western blot analysis using rabbit PPRV H and PPRV F polyclonal antibodies. (**a**) M: protein ladder; 1–6: P3, P6, P9, P12, P15 and P18 generations of recombinant virus rSRV9-H; 7: negative control; (**b**) M: protein ladder; 1–6: 3, 6, 9, 12, 15, and 18 generations of recombinant virus rSRV9-H; 3, 6, 9, 12, 15, and 18 generations of recombinant virus rSRV9-F; 7: negative control.

**Figure 5 vetsci-09-00555-f005:**
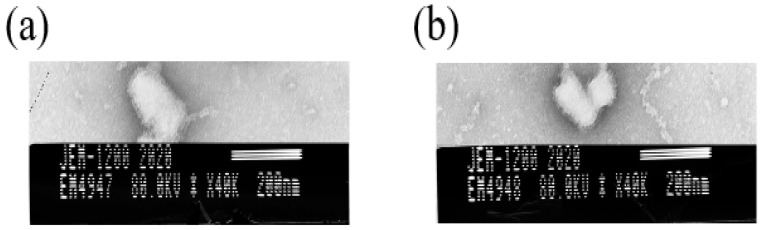
Observation of the recombinant viruses rSRV9-H and rSRV9-F via transmission electron microscopy. The supernatant of recombinant viruses (**a**) rSRV9-H and (**b**) rSRV9-F were inactivated with glutaraldehyde and dropped onto a special copper mesh for TEM. After incubation at room temperature for 5 min, the cells were stained with 1% phosphotungstic acid for 3 min. The excess liquid on the edge of the copper mesh was removed using filter paper. After the copper mesh was dry, it was observed under a transmission electron microscope.

**Figure 6 vetsci-09-00555-f006:**
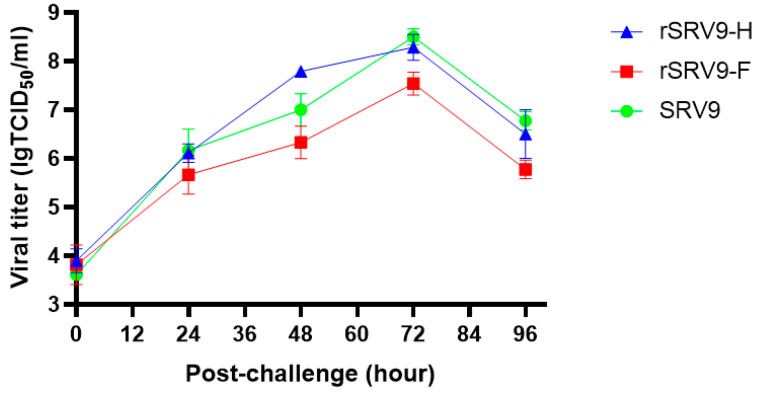
Growth kinetics assay of recombinant viruses rSRV9-H and rSRV9-F. At 0, 24, 48, 72, and 96 h after infection with recombinant viruses, 200 μL of the supernatant was collected, and virus titers were determined by TCID _50_ at 5 timepoints.

**Figure 7 vetsci-09-00555-f007:**
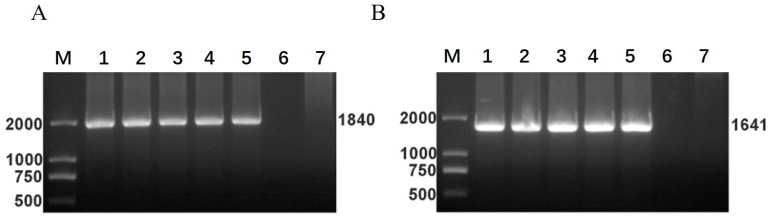
Genetic stability of the recombinant viruses rSRV9-H and rSRV9-F. The P3, P6, P9, P12, and P15 generations of recombinant RABV rSRV9-H and rSRV9-F were used for PCR identification. (**A**) M: DNA marker; 1–5: P3, P6, P9, P12, and P15 generations of recombinant virus rSRV9-H; 7: negative control; (**B**) M: DNA marker; 1–5: P3. P6, P9, P12, and P15 generations of recombinant virus rSRV9-F; 7: negative control.

**Table 1 vetsci-09-00555-t001:** Primers used in this study.

Primer	Sequence (5′–3′)	Enzye
PM/PPRV/H/F	TTTCGTACG*GCCACC*ATGTCCGCACAAAGGGAAAGGATCA	PmeI
PM/PPRV/H/R	ACAGTTTAAACTCAAACTGGATTGCATGTTACCTCT	BsiWI
PM/PPRV/F/F	TTTCGTACG*GCCACC*ATGACACGGGTCGCAATCTTGACAT	PmeI
PM/PPRV/F/R	ACAGTTTAAACCTACAGTGATCTCACATACGACTTT	BsiwI

## Data Availability

The original contributions presented in the study are included in the article. Further inquiries can be directed to the corresponding author.

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
