# Peer review of "Construction of Recombinant Rabies Virus Vectors Expressing H or F Protein of Peste des Petits Ruminants Virus"

_vetsci, 2022, doi:10.3390/vetsci9100555_

Round 1

Author Response

  1. Thank you so much for the professional comment. We are quite agree with you that there is a long way to go to develop vaccines. We also agree with you that It is very necessary to evaluate the vaccine with a mouse model. Actually, we are evaluating of the immune efficacy of recombinant rabies virus vaccines with a mouse model. The preliminary results showed that the prepared inactivated vaccines, i.e., rSRV9-H and rSRV9-F, combined with adjuvant had good immunogenicity and induced Th1/Th2 immune responses in immunized mice. In addition, PPRV- and RABV-neutralizing antibodies were produced, protecting mice from lethal attacks by RABV. The details will be presented in our future papers. We also have added some discussions about our future research direction of the recombinant rabies virus vaccines in the discussion section.
  2.  A reference has been added to the indicated position as shown in line 94.
  3. The H and F proteins are highly conserved, so we chose these two proteins for the developing PPRV vaccine. We agree with you that PPRVs may escape the immunity triggered by our vaccine. We will pay attention on this issue in the future.
  4. Thank you for the interesting comment. There is no data to support that our vaccine own a more heat resistant than other vaccines.
  5. Yes, the haemagglutinin protein (H protein) and the fusion protein (F protein) are Structural genes and are incorporated into the viral particles.
  6. We perform the TCID50 assay by observing the cytopathic effect. We thank you for the
  7. We thank you for your advice. We agree with you that more viral particles will make the result more reliable. However, there is no condition to repeat this experiment. So we ask for your forgiveness. In addition, we will avoid similar mistakes in the future.
  8. Figure 3-1 A should be Figure 1A, we are sorry for the carelessness, and we have corrected this in the revised manuscript.
  9. (be) should be (b-e), we are sorry for the carelessness. We have corrected this in the revised manuscript.
  10. We thank for your careful review. rSRV9-Egfp was not identified here. We have deleted the related description in the revised manuscript.
  11. We have corrected the caption for figure 4b.
  12. We have deleted the repeated content in the revised manuscript.

Reviewer 2 Report

Congratulation for excellent research work done on economically important transboundary animal disease which is considered for eradication after rinderpest! The need of thermostable vaccine in tropical conditions, need of virus neutralization test for sero- monitoring and declaration of PPR free status, DIVA issue for livestock trade are well captured for justification of viral vector vaccine R&D. 

Being rabies expert, it is a critical question that why rabies virus was chosen as a viral vector as it is mainly dog-mediated disease whereas PPR is caprine and ovine specific (host-specific and no scope to use for other animal species). In my opinion, the selection of RABV is a fit for purpose justification. Among livestock, cattle (above 60%) are victim of rabies and not sheep and goats in rabies endemic settings). If we plan to produce bivalent vaccine for Canine distemper and rabies or PPR and sheep or goat pox, it will make a common sense. It is true that viral vector vaccine candidates for veterinary use have been limited for research purpose which is unfortunate, and this vaccine should not be destined for this purpose using rabies virus as a vector! There are successful recombinant rabies vaccines used for oral rabies immunization of wild animals, but reverse genesis and residual pathogenesis are challenging! However, new technological advancement has addressed this issue for use in dog population. Vaccinia virus and other nonpathogenic viral vectors have been used for recombinant rabies vaccine. 

I would suggest to provide concluding remarks (elaborate last paragraph from discussion and way forward such as in vivo or field clinical trial etc.. 

Author Response

Comments 1:Being rabies expert, it is a critical question that why rabies virus was chosen as a viral vector as it is mainly dog-mediated disease whereas PPR is caprine and ovine specific (host-specific and no scope to use for other animal species). In my opinion, the selection of RABV is a fit for purpose justification. Among livestock, cattle (above 60%) are victim of rabies and not sheep and goats in rabies endemic settings). If we plan to produce bivalent vaccine for Canine distemper and rabies or PPR and sheep or goat pox, it will make a common sense. It is true that viral vector vaccine candidates for veterinary use have been limited for research purpose which is unfortunate, and this vaccine should not be destined for this purpose using rabies virus as a vector! There are successful recombinant rabies vaccines used for oral rabies immunization of wild animals, but reverse genesis and residual pathogenesis are challenging! However, new technological advancement has addressed this issue for use in dog population. Vaccinia virus and other nonpathogenic viral vectors have been used for recombinant rabies vaccine. 

Response 1: We are greatly honored to receive your professional and unique perspective comments. We chose to use RABV as our vector for the RVFV vaccine for the following reasons: (1) there are no pre-existing antibodies to RABV, which would affect the use of the vaccine vector; (2) RABV replicates in the cytoplasm so the viral genome does not integrate into the host genome; (3) There is no danger of regaining virulence after RABV virions are inactivated; (4) The RABV genome can stably accommodate large foreign genes.

We quite agree with your advice to produce bivalent vaccine for Canine distemper and rabies or PPR and sheep or goat pox. We thank you for the useful advice, and we will pay more attention to these fields.

We also agree with you that viral vector vaccine candidates for veterinary use have been limited for research purpose. Although the application of viral vector vaccine is far away, exploration is always on the way. We thank you for the tolerance and support.

We are also sorry for the unclear description in the original manuscript, we have added some of the reasons we choose RABV vector in the revised version.

We are well aware that the manuscript has many shortcomings, and we thank you for your kind and patient review. We also hope you will be satisfied with our response.

Comments 2:I would suggest to provide concluding remarks (elaborate last paragraph from discussion and way forward such as in vivo or field clinical trial etc..  some content about the in vivo study in the next study in the discussion section.

Response 2: Thank you so much for the professional comment. We have added some discussions about our future research direction of the recombinant rabies virus vaccines in the discussion section.

Reviewer 3 Report

Authors should includereferences about if RV has been used as vector previously (p2 L62-66)

Fig 7, cut both gels in a proffessional way

Author Response

Authors should include references about if RV has been used as vector previously (p2 L62-66)

Response 1:Thank you for your useful comment. Although few, RVFV has indeed used as vector in vaccine fields. The RVFV vector has been successfully used in developing vaccines against severe fever with thrombocytopenia syndrome (SFTS) [1], bovine ephemeral fever virus[2], Nipah virus[3]. Our lab has also focused on the RV vector studies for years, and has developed a Rift Valley Fever Virus Vaccine with this system [4]. We have added these details in the revised manuscript. The missed references have been also been added.

Fig 7, cut both gels in a proffessional way

Response 2:Thank you for you useful advice, we have cut the gels accordingly and replaced the figure 7.

References:

[1] L. Tian, L. Yan, W. Zheng, X. Lei, Q. Fu, X. Xue, X. Wang, X. Xia, and X. Zheng, A rabies virus vectored severe fever with thrombocytopenia syndrome (SFTS) bivalent candidate vaccine confers protective immune responses in mice. Veterinary microbiology 257 (2021) 109076.

[2] W. Zheng, Z. Zhao, L. Tian, L. Liu, T. Xu, X. Wang, H. He, X. Xia, Y. Zheng, Y. Wei, and X. Zheng, Genetically modified rabies virus vector-based bovine ephemeral fever virus vaccine induces protective immune responses against BEFV and RABV in mice. Transboundary and emerging diseases 68 (2021) 1353-1362.

[3] R. Keshwara, T. Shiels, E. Postnikova, D. Kurup, C. Wirblich, R.F. Johnson, and M.J. Schnell, Rabies-based vaccine induces potent immune responses against Nipah virus. NPJ vaccines 4 (2019) 15.

[4] S. Zhang, M. Hao, N. Feng, H. Jin, F. Yan, H. Chi, H. Wang, Q. Han, J. Wang, G. Wong, B. Liu, J. Wu, Y. Bi, T. Wang, W. Sun, Y. Gao, S. Yang, Y. Zhao, and X. Xia, Genetically Modified Rabies Virus Vector-Based Rift Valley Fever Virus Vaccine is Safe and Induces Efficacious Immune Responses in Mice. Viruses 11 (2019).

Reviewer 4 Report

This paper talks about using a RABV vector to construct clones for PPRV H and F proteins. This is an important topic since the circulating viruses are bringing big risks in husbandry, and it is important to move forward in the vaccine studies in the related field. The overall description and flow of the paper is nice.

However, it could be necessary to discuss more on the potential downsides or risks, or any successful examples using this type of technique in vaccine studies in the related field.

It is also necessary to have animal models which will bring more valuable and reliable information to the study.

Other small points will be references, in the introduction part. Please go over the texts again, especially in introduction part and cite papers after each of your solid statements. 

Author Response

Comment 1: However, it could be necessary to discuss more on the potential downsides or risks, or any successful examples using this type of technique in vaccine studies in the related field.

 Response:Thank you for the professional advice. We added the potential downsides or risks, and some successful examples using RVFV vector in vaccine studies. Although few, RVFV has indeed used as vector in vaccine fields. The RVFV vector has been successfully used in developing vaccines against severe fever with thrombocytopenia syndrome (SFTS)[1], bovine ephemeral fever virus[2], Nipah virus[3]. Our lab has also focused on the RV vector studies for years, and has developed a Rift Valley Fever Virus Vaccine with this system[4]. We have added these details in the revised manuscript.

Comment 2: It is also necessary to have animal models which will bring more valuable and reliable information to the study.

Response: We thank you for the professional advice. Actually, we are evaluating of the immune efficacy of recombinant rabies virus vaccines with a mouse model. The preliminary results showed that the prepared inactivated vaccines, i.e., rSRV9-H and rSRV9-F, combined with adjuvant had good immunogenicity and induced Th1/Th2 immune responses in immunized mice. In addition, PPRV- and RABV-neutralizing antibodies were produced, protecting mice from lethal attacks by RABV. The details will be presented in our future papers.

Comment 3: Other small points will be references, in the introduction part. Please go over the texts again, especially in introduction part and cite papers after each of your solid statements. 

Response:  The missed references have been added in the revised manuscript.

Round 2

Reviewer 1 Report

I appreciate the improvements made in the revised manuscript. However, the main issue, the immunogenicity of the recombinant viruses was not addressed. I understand that the experiments in vivo are currently underway. Presenting even the most preliminary result would greatly improve this manuscript, and should not hamper the publication of the more comprehensive in vivo study in the future. If the Authors feel that it is not possible to include such results, I would ask at least for the experimental evidence that the viral particles contain H and F proteins. The proof for that is still missing and should not be too hard to get.  I would suggest a western blot on samples of the purified virus, with a control blot made with anti-rabies antibody.

What was analyzed in the western blot fig. 4? Cell lysate, medium?

Author Response

Dear reviewer,

We thank you for the time and effort have spent on our manuscript. Your advice is professional and useful.

(1) We agree with you that presenting the result of in vivo study would greatly improve our manuscript. However, we have submitted the related result which entitled “Evaluation of the immune efficacy of recombinant rabies virus vaccines expressing the H and F proteins of peste des petits ruminants in mice” to another journal. So we are sorry that we could not duplicate publication of these results in this study. To reduce your concerns, we provide the abstract of the in vivo study, as following:

Abstract

Peste des petits ruminants (PPR) is one of the most contagious and lethal small ruminant diseases with a global distribution. It is the target disease of a global eradication campaign led by the Office International des Epizooties (OIE) and the Food and Agriculture Organization of the United Nations (FAO). Viral vector vaccines are considered one of the most promising methods for the development of new peste des petits virus (PPRV) vaccines. Previously, using reverse genetics system, our research group successfully constructed recombinant rabies viruses rSRV9-H and rSRV9-F, which express the envelope glycoprotein H (haemagglutinin protein) or F (fusion protein) genes of PPRV, respectively. In this study, 2 recombinant viruses were inactivated and mixed with adjuvants to prepare vaccines, and the immune efficacy of the recombinant viruses was evaluated using a mouse model. The results indicated that the 2 recombinant virus vaccines, rSRV9-H and rSRV9-F, induced different levels of cellular and humoral immune responses in mice, produced effective neutralizing antibodies against PPRV and rabies virus (RABV), and resisted attack by RABV. Notably, the immune performance of recombinant virus rSRV9-H was superior to that of rSRV9-F in all aspects. The results from this study indicate that bivalent inactivated vaccine candidate strains for the prevention of rabies in PPR and livestock rabies are safe and effective, providing new directions for the development of novel PPRV vaccines based on viral vector systems.

 (2) In the manuscript, the recombinant viruses have been identified by direct immunofluorescence, indirect immunofluorescence, as well as Western blot (Figure 4).

(3) The western blot in fig. 4 is tested for the Cell lysate. We are sorry for the unclear description in the material and method section.